# An Ancillary Method for Adrenal Venous Sampling in Cases in Which Right Adrenal Vein Sampling Is Difficult

**DOI:** 10.3390/diagnostics13040649

**Published:** 2023-02-09

**Authors:** Akira Yamamoto, Takeshi Fukunaga, Mitsuru Takeuchi, Hiroki Nakamura, Akihiko Kanki, Atsushi Higaki, Tsutomu Tamada

**Affiliations:** 1Department of Radiology, Kawasaki Medical School, Kurashiki 701-0192, Japan; 2Department of Radiology, Radiolonet Tokai, Nagoya 460-8501, Japan

**Keywords:** adrenal venous sampling, alternative methods, laterality index, modified laterality index

## Abstract

Catheterization of the right adrenal vein (rt.AdV) to obtain blood samples can often be difficult. The aim of the present study was to investigate whether blood sampling from the inferior vena cava (IVC) at its juncture with the rt.AdV can be an ancillary to sampling of blood directly from the rt.AdV. This study included 44 patients diagnosed with primary aldosteronism (PA) in whom AVS with adrenocorticotropic hormone (ACTH) was performed, resulting in a diagnosis of idiopathic hyperaldosteronism (IHA) (*n* = 24), and patients diagnosed with unilateral aldosterone-producing adenoma (APA) (*n* = 20; rt.APA = 8, lt.APA = 12). In addition to regular blood sampling, blood was also sampled from the IVC, as the substitute rt.AdV [S-rt.AdV]. Diagnostic performance with the conventional lateralized index (LI) and the modified LI using the S-rt.AdV were compared to examine the utility of the modified LI. The modified LI of the rt.APA (0.4 ± 0.4) was significantly lower than those of the IHA (1.4 ± 0.7) (*p* < 0.001) and the lt.APA (3.5 ± 2.0) (*p* < 0.001). The modified LI of the lt.APA was significantly higher than those of the IHA (*p* < 0.001) and rt.APA (*p* < 0.001). Likelihood ratios to diagnose rt.APA and lt.APA using the modified LI with threshold values of 0.3 and 3.1 were 27.0, and 18.6, respectively. The modified LI has the potential to be an ancillary method for rt.AdV sampling in cases in which rt.AdV sampling is difficult. Obtaining the modified LI is extremely simple, which might complement conventional AVS.

## 1. Introduction

Primary aldosteronism (PA) is recognized as the most common cause of secondary hypertension. Current diagnostic approaches have shown that the prevalence of PA in patients with hypertension is around 10%. This figure is much higher than previously estimated, and PA is believed to account for 17%–23% of cases of refractory hypertension [1,2]. When selecting the treatment strategy for PA, the localization of hormone-producing lesions is crucial, because a unilateral aldosterone-producing adenoma (UAPA) can be treated by laparoscopic adrenalectomy or open surgery, which may result in a cure or significant improvement of blood pressure, thus decreasing the incidence of cardiovascular and cerebrovascular complications [1,3,4,5]. However, drug treatment is routinely chosen for idiopathic hyperaldosteronism (IHA), which is caused by bilateral adrenal glands.

Various noninvasive examinations, such as computed tomography (CT), magnetic resonance imaging (MRI), and iodocholesterol (NP-59) scintigraphy performed under dexamethasone suppression, have been used to distinguish UAPA from IHA. However, CT/MRI misdiagnosed the subtyping of PA in 37.8% of patients [6]. On nuclear imaging, the reported diagnostic accuracy of the criterion for diagnosing the laterality of aldosterone secretion was only 47% [7]. These imaging examinations were unreliable to confirm surgically curable types and were not suitable for managing patients with PA properly. AVS may be replaced by 11C-metomidate PET in the future but has not yet shown sufficient results to demonstrate its usefulness over AVS. The issue of the half-life of the nuclide should also be considered [8].

Currently, adrenal venous sampling (AVS) serves as the gold standard for subtyping of PA, with the highest diagnostic accuracy of any test [1,9]. However, AVS has several problems. These include its invasiveness compared with other imaging modalities, as well as the potential for bleeding, infection, vascular damage, and other complications associated with catheterization. However, the most important factor is that the technical success rate of AVS is differs widely among facilities. The reported success rates of AVS range from 55% to 98%, with technical success rates varying widely depending on the operator’s skill with the right adrenal vein (rt.AdV) sampling [1,10,11]. Right AdV catheterization is often difficult due to unfavorable anatomy of the vein, including small vein size, short length, caudal direction for the transfemoral approach, and rare drainage to the accessory hepatic vein. Even in patients with successful right AdV catheterization, the catheter might get dislodged with respiration, or the tip of the catheter might become lodged in the vascular wall, making blood sampling difficult. All of these factors limit the widespread use of this approach [1,3,10,11,12,13,14]. Given the wide range of success rates for right AVS [10,11,15], it is necessary to achieve a more consistent AVS success rate and eliminate the technical failures associated with right AVS cannulation [16].

Identification of the confluence of the right AdV with the inferior vena cava (IVC) by preoperative contrast-enhanced thin-slice computed tomography (CT) is an essential and feasible procedure [17]. Catheterization of the right AdV is often difficult, but if preoperative CT is used as a reference, the catheter tip can very easily be guided downstream to its confluence with the IVC.

The aim of the present study was to investigate whether blood sampling from the IVC at its juncture with the right AdV can be an ancillary method to sampling blood directly from the right AdV.

## 2. Materials and Methods

### 2.1. Patient Population

This study was conducted in accordance with the Declaration of Helsinki and approved by the Ethics Committee of Affiliated Hospital of Kawasaki Medical School (Kurashiki, Japan; ID No. 3288). As it was a retrospective study, the requirement for informed consent was waived by the Ethics Committee of Affiliated Hospital of Kawasaki Medical School (Kurashiki, Japan; ID No. 3288). This study initially included 58 consecutive patients who underwent AVS with adrenocorticotropic hormone (ACTH) for local diagnosis of the lesion because of suspected primary aldosteronism (PA) with hypertension and an aldosterone-renin ratio (ARR = plasma aldosterone concentration, i.e., PAC)-plasma renin activity (PRA) ratio: (PAC/PRA)) > 200 [9] from January 2010 to February 2019. The following patients were excluded because AVS could not be technically completed or the final local diagnosis of the hormone-producing lesion was unclear: patients who could not complete blood sampling from the right AdV (*n* = 9); patients who did not undergo surgery, despite being diagnosed with unilateral aldosterone-producing adenoma (UAPA) by AVS (*n* = 2); patients whose ARR did not decrease to within the normal range (<200) after surgery (*n* = 2); patients whose blood pressure did not decrease to within the normal range (<140/90) after surgery (*n* = 0); patients whose postoperative course could not be followed up (*n* = 1). Finally, this study included 44 patients (21 men, 23 women; mean age 56.0 years, range 28–73 years) diagnosed by AVS, and all patients who were diagnosed with UAPA by AVS who underwent surgery and whose ARR decreased to the normal range (<200). Preoperative CT showed adrenal lesions in 30 cases and no lesions in 14 cases which were diagnosed by a radiologist. The lesion localization of 30 cases with adrenal gland was 9 cases in the right adrenal gland, 17 cases in the left adrenal gland, and 4 cases in the bilateral adrenal gland. The resulting diagnosis was idiopathic hyperaldosteronism (IHA) (*n* = 24) and unilateral aldosterone-producing adenoma (UAPA) (*n* = 20) including right APA (*n* = 8) and left APA (*n* = 12). Participants’ demographics and relevant clinical details are shown in Table 1. 

### 2.2. AVS Procedure

AVS was always conducted by 2 or 3 senior radiology residents or a radiologist with supervision by one of four interventional radiologists, after the administration of anticoagulant therapy with 3000 units of heparin. Under local anesthesia, 4-Fr sheaths were percutaneously inserted in the femoral veins of both legs. Before and after (between 15 and 30 min) bolus injection of 0.20 mg of the adrenocorticotropic hormone (ACTH) (cosyntropin), at least 3 mL of venous blood were sampled from the following four sites: the lowest point of the inferior vena cava (IVC) and the left AdV, the right AdV, and the IVC at one vertebral level above its confluence with the right AdV (the substitute right AdV [S-rt. AdV]). For the S-rt. AdV, the tip of the catheter was positioned on the same side of the IVC as the right AdV. Figure 1 shows how the positions of the S-rt. AdV were determined. For cannulation of the left AdV, a 4-Fr catheter of Simons-type (Terumo, Tokyo, Japan) and a 1.7-Fr microcatheter (Boston Scientific, Marlborough, MA, USA) were mainly used, and sampling from the main trunk of the left AdV was conducted via a 4-Fr sheath (Medikit, Tokyo, Japan) inserted into the left femoral vein. For cannulation of the right AdV, a 4-Fr Cobra-type (Medikit, Tokyo, Japan) or Hook-type (Medikit, Tokyo, Japan) catheter inserted into the right femoral vein via a 4-Fr sheath was used. The same microcatheter was used for the right AdV if blood could not be withdrawn from the 4-Fr catheter. Digital subtraction angiography (DSA) was performed using an Allura Xper FD20 (Philips Healthcare, Best, Netherland) and a Infinix 8000v (Canon Medical Systems, Tochigi, Japan). For patients in whom it was difficult to determine whether the catheter had entered the right AdV, an unenhanced or enhanced cone-beam CT was performed to confirm the locations of the catheters in the right AdV. 

## 3. Data Analysis

To evaluate successful catheterization of an AdV, the selectivity index (SI) was calculated as follows: SI = plasma cortisol concentrations of the AdV/plasma cortisol concentrations of the lowest point of the IVC. On the basis of guidelines in previous reports [18,19,20], an SI ≥ 5.0 was considered to indicate successful catheterization of each AdV.

The aldosterone/cortisol (A/C) ratio of each AdV was calculated to evaluate the hormone-producing local diagnostic ability for PA. The conventional lateralized index (LI) (A/C ratio larger)/(A/C ratio smaller) was calculated. On the basis of guidelines in previous reports (2022), an LI < 4.0 was diagnosed as IHA, and an LI ≥ 4.0 was diagnosed as UAPA. Our own modified lateralized index (modified LI) (Left AdV A/C ratio)/(S-rt.AdV A/C ratio) was also calculated, the cutoff value with the best diagnostic performance was determined, and the diagnostic performances of the conventional LI and modified LI were compared.

### 3.1. Principle of Local Diagnosis of PA by the Modified LI

To explain the modified LI in simple terms, assume that a hormone-producing lesion produces 10 units of hormones, and a normal adrenal gland produces one unit. Taking no account of hormone dilution, if there is a hormone-producing lesion in the right adrenal gland, then the one unit secreted normally and the 10 units of oversecreted hormone produced by the lesion in that gland, making a total of 11 units, will be detected in the blood sampled from the S-rt. AdV (Figure 2a). If there are hormone-producing lesions in both adrenal glands, the 10 units oversecreted by the hormone-producing lesion in the left adrenal gland and the 10 units oversecreted by the hormone-producing lesion in the right adrenal gland, making a total of 20 units, will be detected (Figure 2b). If there is a hormone-producing lesion in the left adrenal gland, the 10 units oversecreted by the hormone-producing lesion in the left adrenal gland and the one unit secreted by the right adrenal gland, a total of 11 units, will be detected (Figure 2c). Substituting these values into the formula for calculating the modified LI [modified LI = (Lt. AdV A/C ratio)/(S-rt.AdV A/C ratio) gives the following results. If there is a hormone-producing lesion in the right adrenal gland (Rt. UAPA), the modified LI = 1/11 = 0.1 (Figure 2a). If there are hormone-producing lesions in both adrenal glands (IHA), the modified LI = 10/20 = 0.5 (Figure 2b). If there is a hormone-producing lesion in the left adrenal gland (Lt.UAPA), the modified LI = 10/11 = 0.9 (Figure 2c). Thus, the results differ among these three cases. In fact the hormone is diluted in the bloodstream, but the location of the hormone-producing lesion(s) will nevertheless make some difference.

### 3.2. Statistical Analysis

A nonparametric Mann–Whitney U test was used to compare and identify significant differences in the conventional LI and the modified LI between UAPA and IHA. A receiver operating characteristic (ROC) curve analysis was performed for the comparison of UAPA and IHA using the conventional LI and the modified LI. The area under the ROC curve (AUC) was calculated for each variable, and the optimal threshold for each variable was determined from its respective ROC analysis by evaluating the likelihood ratios (sensitivity/(1-specificity)) at different cutoff points on the ROC curve. The cutoff values of the modified LI values were determined so that the likelihood ratios were maximized. All statistical analyses were performed using SPSS software (v. 17.0J for Windows; Chicago, IL, USA). All tests were two-sided, and *p* < 0.05 was considered significant.

## 4. Results

In a comparison among rt. APA, lt. APA, and IHA, the ARR of pre-operation were 1429.9 ± 902.6, 1562.1 ± 1429.9 and 407.6 ± 356.8, respectively. The ARR of post-operation were 90.7 ± 55.0 after right adrenal gland resection and 89.6 ± 44.0 after left adrenal resection (Table 2). Conventional LIs of right APA (23.3 ± 25.8) and left APA (7.0 ± 5.3) were significantly higher than that of IHA (1.8 ± 2.5) (*p* = 0.003 and *p* < 0.001) (Figure 3). The modified LI of right APA (0.4 ± 0.4) was significantly lower than those of IHA (1.4 ± 0.7) (*p* < 0.001) and left APA (3.5 ± 2.0) (*p* < 0.001). The modified LI of left APA was significantly higher than those of IHA (*p* < 0.001) and right APA (*p* < 0.001) (Figure 4, Table 2).

On ROC curve analysis for the diagnostic performance of the conventional LI, the area under the curve (AUC) was 0.90 in UAPA, whereas for the modified LI, the AUC was 0.92 in the right APA and 0.81 in the left APA (Figure 5). Likelihood ratios to diagnose UAPA using the conventional LI were 15.6, with a threshold value of 3.4, and to diagnose the right APA and left APA using the modified LI, they were 27.0, and 18.6, respectively, with threshold values of 0.3 and 3.1, respectively (Table 3).

In five of the 44 cases in which catheterization of the right AdV was deemed technically successful, the SI was <5.0, suggesting unsuccessful catheterization. For the left AdV, however, the SI was >5.0 in all cases, indicating successful catheterization. For the five cases of suspected unsuccessful right AdV catheterization, the localization of the hormone-producing lesion according to the conventional LI was mistaken in three. Localization by the modified LI was mistaken in only one case.

## 5. Discussion

In this study, the diagnostic performance of the modified LI to identify the location of hormone-producing lesions was not much inferior to the diagnostic performance of the conventional LI. One reason for this may be that, in many cases, the ACTH challenge greatly increases aldosterone secretion, such that its diagnostic performance is comparatively good even when blood is sampled from the IVC, where the aldosterone concentration is diluted, instead of from the right AdV. According to a review of 47 reports, the success rate of cannulating the right AdV in 384 patients was 74% [1,15], but with experience, the success rate increases to 90–96% [1,10,19]. As shown by that result, the technical success rate of AdV sampling is thus highly dependent on the operator’s skill. However, the demand for AdV sampling in recent years makes it an investigation that should be provided by more hospitals, and it is undesirable that the results be affected by the operator’s skill. In most cases, unsuccessful AdV sampling is due to either unsuccessful catheterization of the right AdV or an inability to suction blood from this vein despite its successful catheterization. 

Previously, some studies reported the diagnostic utility of PA from adrenal venous sampling without the right adrenal vein [21,22,23]. However, these papers are based on the results of blood sampling from the left adrenal vein and IVC at the lower point of the confluence of bilateral renal vein. The decisive difference in our study is that our IVC blood sampling is taken from the upper point of the confluence of the right AdV, whereas the previous studies are from the IVC at the lower point of the confluence of the bilateral renal vein. Therefore, previous studies have evaluated the changes that rely on suppression of hormone secretion from the left adrenal gland in case of a hormone producing lesion in the right adrenal gland. On the other hand, in our study, although it is diluted by IVC flow, hormone secretion from the right adrenal lesion is added to the result, so it seems that the diagnostic accuracy is further improved.

Assessment using our modified LI may resolve this problem by making the selection of the right AdV unnecessary, instead using the IVC where catheter placement is extremely simple. Although selecting and sampling from the right AdV are indisputably the best, when this cannot be achieved, sampling from the IVC at one vertebral level above its confluence with the right AdV (the S-rt. AdV) can be performed simply and reliably. Assessment using the modified LI may then provide data on which the treatment strategy can be based.

Even more interestingly, for the five cases in which catheterization of the right AdV was judged to have been technically successful, but the SI data indicated unsuccessful catheterization, assessment using the conventional LI resulted in mistaken localization in 3/5 cases, whereas assessment using the modified LI resulted in mistaken localization in only 1/5 cases, and correctly diagnosed two cases in which the conventional LI provided the wrong localization. This result suggests that the S-rt. AdV should be used not only when right AdV catheterization is technically difficult, but also for patients in whom it has been successfully achieved, since this practice may assist with the localization of hormone-producing lesions in patients with postoperative SI < 5.0. However, the modified LI also has problems. The most important of these is that the position of the catheter for sampling from the S-rt. AdV cannot be determined if the confluence of the right AdV has not been identified intraoperatively. This problem can be overcome by careful evaluation of the confluence of the right AdV on preoperative CT [24]. Furthermore, how the results of modified LI should be reflected in clinical treatment strategies should be thoroughly considered. In the future, this discussion is essential to establish the modified LI as a useful test and to make it a useful test for patients.

A limitation of this study was that the catheter tip for S-rt. AdV blood sampling was positioned after the confluence of the right AdV had been identified intraoperatively. In patients in whom the confluence of the AdV cannot be identified intraoperatively, catheter tip positioning must be entirely dependent on information obtained from preoperative CT, and this might lead to different results. However, careful identification of the right AdV branch with reference to preoperative CT should enable the catheter tip to be positioned with reasonable accuracy. The second limitation of the present study was that cases diagnosed with IHA have not been definitively diagnosed. IHA was not indicated for surgery and could not solve this problem. However, there was no significant difference in diagnostic ability when compared with conventional LI. The third limitation was that the procedures were performed by different operators, which may have caused technical differences. However, since all procedures were conducted under the supervision of an interventional radiologist, these technical differences were probably minor. The fourth limitation was that this study was based on the hypothesis shown in Figure 2, but there is no report to support it. It also matters, of course, as the study aims to demonstrate a new methodology. We believe that it will show the tendency shown in the hypothesis. The fifth limitation was the small number of subjects. Further studies of more cases are required.

In conclusion, the modified LI has the potential to be an ancillary method for right AdV sampling in cases in which right AdV sampling is difficult. Obtaining the modified LI is extremely simple, and it might complement conventional AVS. The modified LI using blood sampled from the IVC at the juncture of the right AdV, which can be done easily in such patients, is a potentially useful clinical method. Further investigation with an increased number of cases is necessary in the future.

## Figures and Tables

**Figure 1 diagnostics-13-00649-f001:**
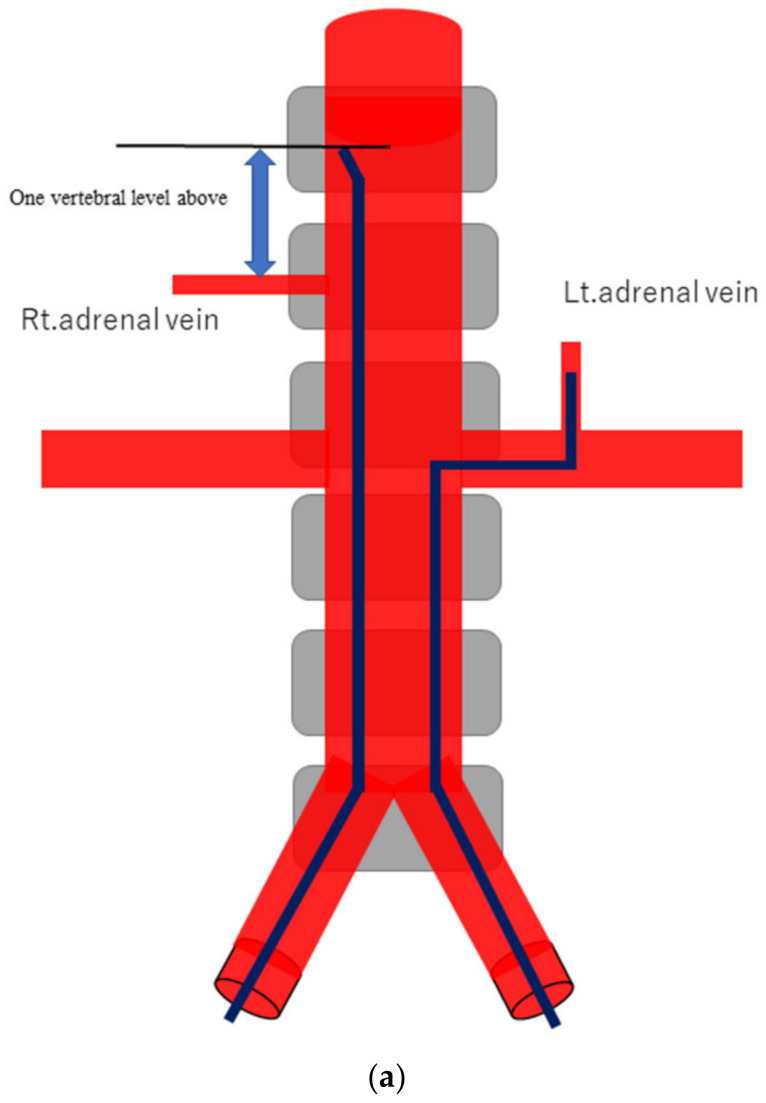
Schematic diagram (**a**) and digital angiography (**b**,**c**) of the location of catheter placement in the S-rt. AdV. The catheter tip is placed at one vertebral level above the confluence of the IVC with the right AdV, and its orientation is adjusted so that it is pointing toward the right dorsal side in the same way as the right AdV.

**Figure 2 diagnostics-13-00649-f002:**
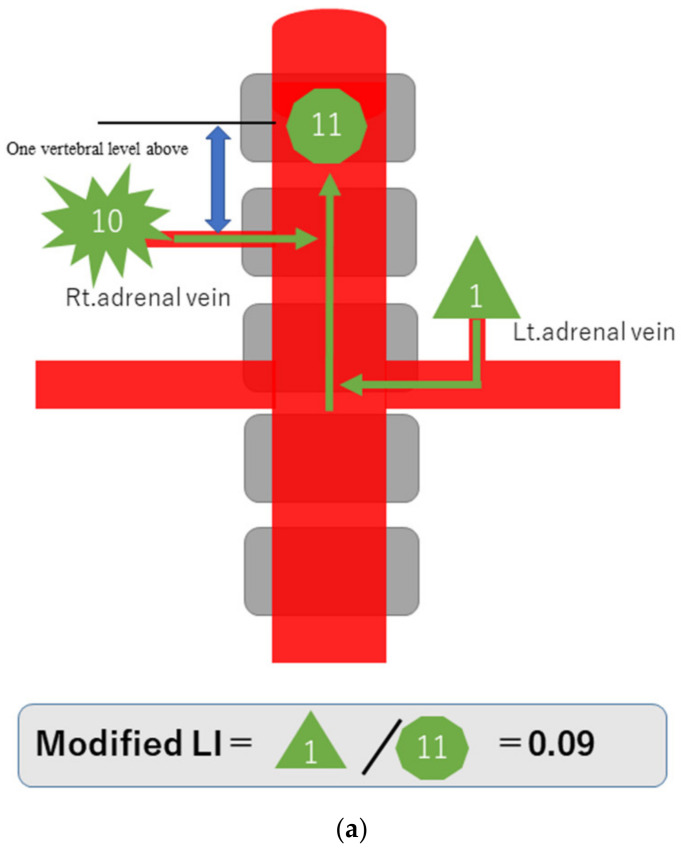
Schematic diagram in case of hormone-producing lesion in the right adrenal gland (**a**), hormone-producing lesions in both adrenal glands (**b**) and hormone-producing lesion in the left adrenal gland (**c**). To explain the modified LI in simple terms, assume that a hormone-producing lesion produces 10 units of hormones, and a normal adrenal gland produces 1 unit. Taking no account of hormone dilution. In case of (**a**), total of 11 units of hormone will be detected in the blood sampled from the S-rt. AdV by the 10 units of oversecreted by the lesion in right adrenal gland and the 1 unit secreted normal left adrenal gland. In case of (**b**), total of 20 units of hormone will be detected in the blood sampled from the S-rt. AdV by the 10 unit oversecreted by the lesion in both adrenal glands. In case of (**c**), total of 11 units of hormone will be detected in the blood sampled from the S-rt. AdV by the one unit secreted normal left adrenal gland and the 10 units of oversecreted by the lesion in right adrenal gland. Substituting these values into the formula for calculating the modified LI, in case of (**a**) (Rt. UAPA), the modified LI = 1/11 = 0.1, in case of (**b**) (IHA), the modified LI = 10/20 = 0.5 and in case of (**c**) (Lt. UAPA), the modified LI = 10/11 = 0.9.

**Figure 3 diagnostics-13-00649-f003:**
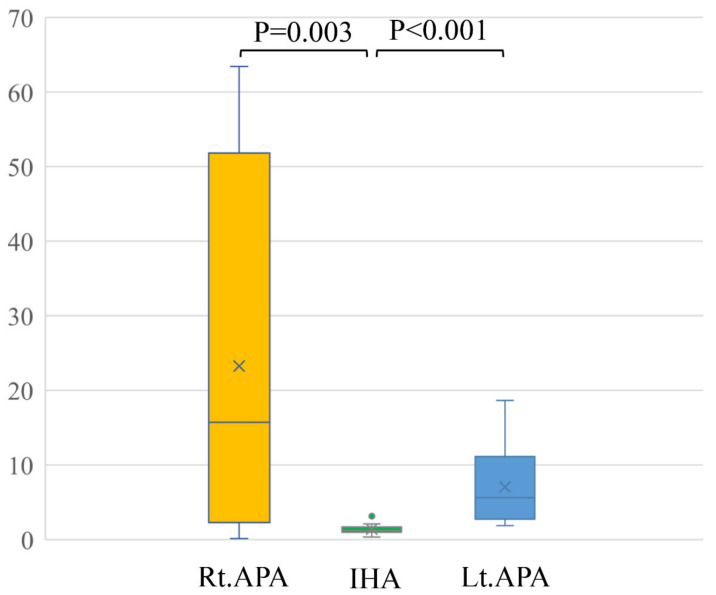
Conventional LIs of right APA (23.3 ± 25.8) and left APA (7.0 ± 5.3) were significantly higher than that of IHA (1.8 ± 2.5) (*p* = 0.003 and *p* < 0.001).

**Figure 4 diagnostics-13-00649-f004:**
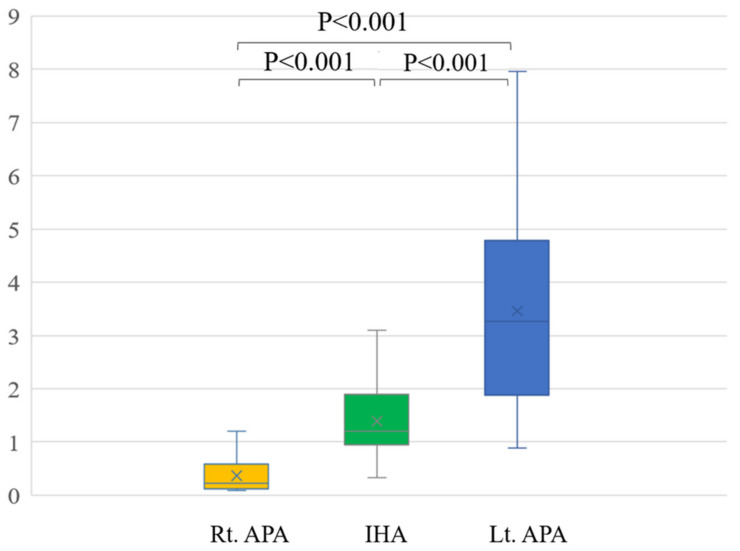
The modified LI of right APA (0.4 ± 0.4) was significantly lower than those of IHA (1.4 ± 0.7) (*p* < 0.001) and left APA (3.5 ± 2.0) (*p* < 0.001). The modified LI of left APA was significantly higher than those of IHA (*p* < 0.001) and right APA (*p* < 0.001).

**Figure 5 diagnostics-13-00649-f005:**
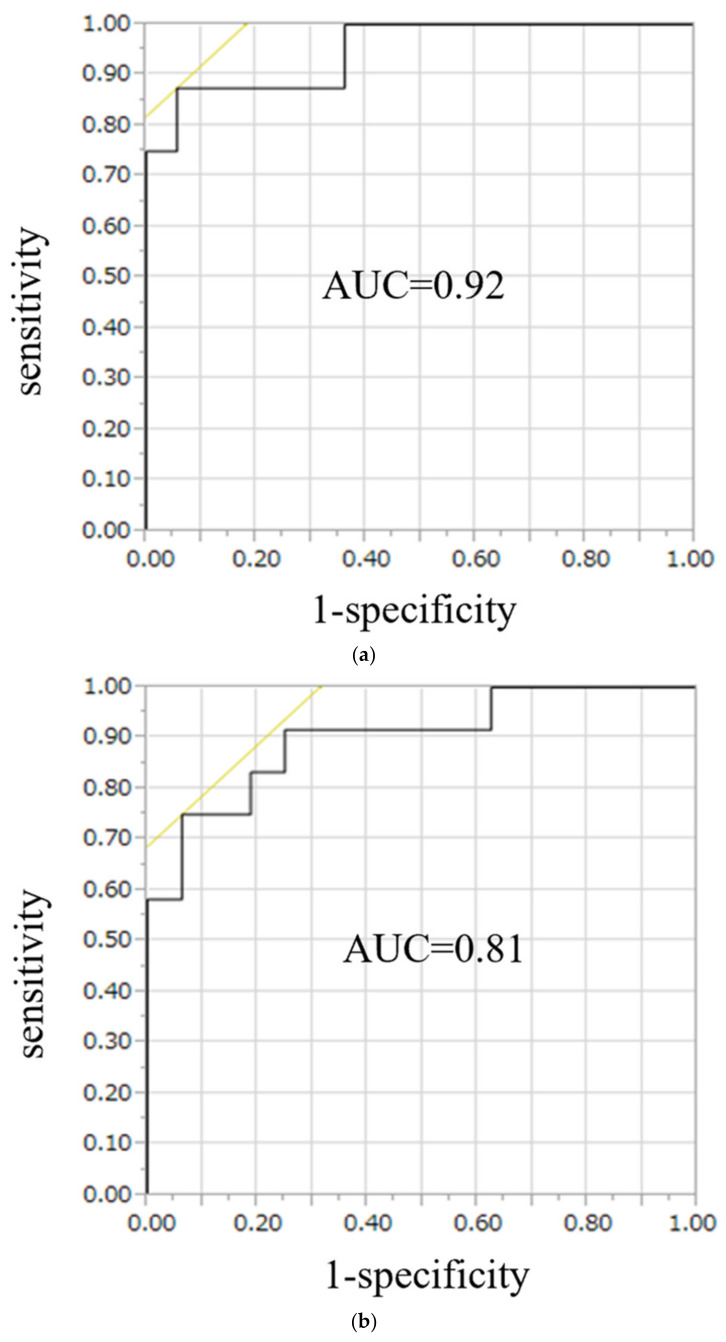
ROC curve analysis for the diagnostic performance of the modified LI, the AUC was 0.92 in right APA (**a**) and 0.81 in left APA (**b**). To diagnose right APA and left APA using the modified LI, likelihood were 27.0, and 18.6, respectively, with threshold values of 0.7 and 2.2, respectively.

**Table 1 diagnostics-13-00649-t001:** 44 participants’ demographics and relevant clinical details.

Characteristics	
Age (year)	Mean: 56 (range: 28–73)
SEX	
Male	21
Female	23
Blood pressure(mmHg)	
Systolic	Mean: 150 (range: 222–110)
Diastolic	91 (130–52)
Antihypertensive drugs at first visit	44/44
PAC (pg/mL) (Mean ± SD)	253.3 ± 255.0
PRA (ng/mL/hr) (Mean ± SD)	0.3 ± 0.2
ARR (Mean ± SD)	992.3 ± 977.0
Adrenal lesion on CT	
No	14
Yes	30
Right	9
Left	17
Bilateral	4

**Table 2 diagnostics-13-00649-t002:** The results of rt. APA vs. Lt. APA vs. IHA.

	UAPA (*n* = 20)		IHA (*n* = 24)
	Right (*n* = 8)	Left (*n* = 12)	
ARR			
Pre-operation (Mean ± SD)	1429.9 ± 902.6	1562.1 ± 1429.9	560.6 ± 356.8
Post-operation (Mean ± SD)	90.7 ± 55.0	89.6 ± 44.0	N/A
Adrenal lesion on CT			
No	1	0	13
Yes	7	12	11
Right	7	0	2
Left	0	10	7
Bilateral	0	2	2
Conventional LI (Mean ± SD)	23.26 ± 25.76	7.05 ± 5.33	1.56 ± 1.32
Modified LI (Mean ± SD)	0.37 ± 0.38	3.46 ± 1.97	1.52 ± 1.11

**Table 3 diagnostics-13-00649-t003:** Diagnostic performance for conventional LI and modified LI.

		AUC	Threshold	Sensitivity	Specificity
Conventional LI	UAPA	0.90	1.9	95%	83%
Modified LI	Rt.APA	0.92	0.7	87%	94%
	Lt.APA	0.81	2.2	75%	94%

## Data Availability

The data presented in this study are available on request from the corresponding author.

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
