# Peer review of "An Ancillary Method for Adrenal Venous Sampling in Cases in Which Right Adrenal Vein Sampling Is Difficult"

_diagnostics, 2023, doi:10.3390/diagnostics13040649_

Round 1
Reviewer 1 Report
The idea of accomplishing diagnosis of primary aldosteronism through a “modified Lateralization Index” based on blood sampling at the site where the inferior vena cava meets the right adrenal vein is catchy, given that many AVS studies failed to achieve bilateral selectivity.
Unfortunately, this is a retrospective observational study which is far too small to provide solid information in this area
I have the following general comment.
I struggled to find the rational for using blood that at best only partially derives from the right adrenal gland. Many centers that experience difficulties in getting selective samples on the right use make side holes on the catheter’s tip. This results into dilution of adrenal blood from the IVC. Moreover, others used the intraprocedural rapid cortisol assay or markers as androstenedione that have a higher step up between the adrenal veins and the IVC. These strategies were neither used as comparison nor discussed.
Specific criticisms.
1. Introduction. Lines 37-43: C14-metomidate PET-CT cannot be disregarded when mentioning the noninvasive localization procedures.
2. Lines 46-49: the complications and invasiveness of AVS are overemphasized and not in line with recent publications as, for example, the AVIS-2 study.
3. Material and Methods, AVS procedure. It is described how the catheterization was performed in four sites. It is specified that the catheter’s tip was positioned on the same side of the IVC as the right AdV at one vertebral level above its confluence with the right AdV. Do the authors suppose that the orientation of the catheter can prevent the blood elution from the AdV into the IVC?
4. Furthermore, the authors add in the limitations that in some patients (how many?) the right AdV confluence was not found intraprocedurally and the catheter’s placement was presumptively based on previous imaging (CT scan). These uncertain data should not be used for validation purposes.
5. On lines 144-177 the authors presented an elaborated theoretically discussion that basically ignores Fick’ principle and the already proposed lateralization Index, Contralateral suppression Index and Relative Aldosterone secretion index.
6. Table 1. Renin is expressed as plasma renin activity, but PRA values is outdated, as the Direct Renin Concentration assay is currently used in most of the civilized world.
7. Results (Lines 215-220). The description of the success rate of the right AdV catheterization and selectivity is unclear and confused.
8. Figure 5, Legend. The legend is too long and not clearly written. Is there a missing full stop in the middle of the sentence?
9. Discussion (lines 244 and 246). There are errors in syntax and spelling.
10. References. Several relevant references are overlooked.
Author Response
Reviewer 1
The idea of accomplishing diagnosis of primary aldosteronism through a “modified Lateralization Index” based on blood sampling at the site where the inferior vena cava meets the right adrenal vein is catchy, given that many AVS studies failed to achieve bilateral selectivity.
Unfortunately, this is a retrospective observational study which is far too small to provide solid information in this area
I have the following general comment.
I struggled to find the rational for using blood that at best only partially derives from the right adrenal gland. Many centers that experience difficulties in getting selective samples on the right use make side holes on the catheter’s tip. This results into dilution of adrenal blood from the IVC. Moreover, others used the intraprocedural rapid cortisol assay or markers as androstenedione that have a higher step up between the adrenal veins and the IVC. These strategies were neither used as comparison nor discussed.
This paper has been checked by native speaker.
Thank you for your comment.
The purpose of this research was devised with the idea of a backup method for the conventional method, not a method to replace the conventional method.
The reason for setting the position of the catheter above one vertebra is to prevent it from being positioned below the right adrenal sinus.
Specific criticisms.
1. Introduction. Lines 37-43: C14-metomidate PET-CT cannot be disregarded when mentioning the noninvasive localization procedures.
We added following sentence in Introduction and reference (8) in Reference
“11C-metomidate PET may replace AVS in the future, but has not yet shown sufficient results to demonstrate its usefulness over AVS. There is also the issue of the half-life of the nuclide (8).”
8. Chen Cardenas SM, Santhanam P. 11C-metomidate PET in the diagnosis of adrenal masses and primary aldosteronism: a review of the literature.
2. Lines 46-49: the complications and invasiveness of AVS are overemphasized and not in line with recent publications as, for example, the AVIS-2 study.
I don't think the content of this paper is significantly different from the results of the AVIS-2 study.
However, we changed or deleted a few sentences as following.
“AVS has a number of number of problems” → “AVS has a number of several problems“
“These include its greater invasiveness compared with other imaging modalities,….”
“…the basic requirement for hospital admission” Because several facilities offer day testing.
We added AVIS-2 study in the reference (17).
3. Material and Methods, AVS procedure. It is described how the catheterization was performed in four sites. It is specified that the catheter’s tip was positioned on the same side of the IVC as the right AdV at one vertebral level above its confluence with the right AdV. Do the authors suppose that the orientation of the catheter can prevent the blood elution from the AdV into the IVC?
We believe that elution into the IVC cannot be prevented, but can be reduced compared to directing it to the opposite side. We also thought that a more rigorous evaluation could be made if the study was conducted in a certain direction.
4. Furthermore, the authors add in the limitations that in some patients (how many?) the right AdV confluence was not found intraprocedurally and the catheter’s placement was presumptively based on previous imaging (CT scan). These uncertain data should not be used for validation purposes.
In this study, rt. AdV was confirmed in all cases and making catetel posision. The Limitation means that catheter position was determined after confirming rt AdV in this study, but if this method is used alone, catheter position should be determined by preoperative CT.
5. On lines 144-177 the authors presented an elaborated theoretically discussion that basically ignores Fick’ principle and the already proposed lateralization Index, Contralateral suppression Index and Relative Aldosterone secretion index.
The main purpose of this paper is to devise a new index that is different from these indices. This is because the procedures for evaluating these indices are cumbersome and often difficult.
6. Table 1. Renin is expressed as plasma renin activity, but PRA values is outdated, as the Direct Renin Concentration assay is currently used in most of the civilized world.
Due to the length of the study, we are evaluating it in PRA; if it is better to add it to Limitation, we will do so.
7. Results (Lines 215-220). The description of the success rate of the right AdV catheterization and selectivity is unclear and confused.
This paragraph means that technical success was not the same as success by SI, and all 44 cases were successful by judgment during the examination, but SI was less than 5.0 in 5 cases.
8. Figure 5, Legend. The legend is too long and not clearly written. Is there a missing full stop in the middle of the sentence?
Thank you. We added “.” In the middle of the sentence.
9. Discussion (lines 244 and 246). There are errors in syntax and spelling.
Thank you. We properly collected “However,” and “sampling”.
10. References. Several relevant references are overlooked.
We added a few references 8),17).
Reviewer 2 Report
The method is interesting as well as the results, however, the protocol to sample from the confluence with the right AdV is not practical. As the author mentioned in the discussion, we will not need to sample from the confluence if we have success to detect the right adrenal vein. They should have sampled from the point that might be around the right adrenal vein considering CT and analyzed it as well.
The theory is simple, but not seem to be persuasive. The concentration gradients of aldosterone and cortisol vary from site to site, partly due to the metabolic effects of the tissues. In the author’s theory, the point for sampling would not have had to be strictly just over one vertebra as long as it is at least above the right adrenal vein. The same might be true if blood is drawn in the left renal vein rather than the left adrenal vein, for example. Unlike the adrenal vein, as the inferior vena cava is large in diameter, its values are likely to be variable because of the mixture of blood flow from many tissues.
Considering those, rather than the values of Modified LI itself being important, the novelty could lie in the fact that this kind of method is expected to gather data, adding new insights to the AVS.
Comments)
In the material and methods section, as primary aldosteronism is sometimes complicated by subclinical Cushing syndrome, the author should describe whether these patients were tested 1 mg dexamethasone suppression test, which could affect the result to analyze LI.
Line 82
If the lateralization in these patients is diagnosed by using modified LI, which side will they be on?
Line 86-87
The sentence “patients whose blood pressure did not decrease to within the normal range (n=0)” is not needed.
Line90-91
Does the normal range mean ARR<200?
Line 91
What is the definition of adrenal lesions? Is the tumor over 10 mm?
Line 96-97
The sentence “At preoperative CT, the adrenal lesions were detected in 30 patients (right: 9, 96 left: 17, bilateral: 4) and 14 patients in which no lesions were detected.” could be omitted being similar to the sentence in line 92-93.
Line 132
I think the correct sentence is “LI < 4.0 was diagnosed as IHA”. However, I can see the patient with LI >4 in the IHA group as an outlier in figure 3. How does the author diagnose this patient with IHA? According to the exclusion criteria, the author seems to consider the postoperative results for definition as well, especially in APAs.
Line 188
Considering the variation of SD in Tables, I recommend the author use median and quartiles and perform a Mann-Whitney test when checking the significance.
Line 191
Considering the values in Table 2, it seems to be a mistake for 1142.9 instead of 11429.9.
Line 208
The purpose to perform AVS is to find the unilateral lesion, meaning the specificity to detect it is important. The author might want to describe the values showing 100% specificity.
Line 215-216
It is said that in 5 of the 44 cases in which catheterization of the right AdV was deemed technically successful, the SI was<5, suggesting unsuccessful catheterization. In general, I think it is a technical error if they cannot get the result that SI was >5. Does this sentence mean these patients have nonfunctional adrenal adenomas having the latent secretion of cortisol or some veins to the adrenal vein, affecting the AVS results? Anyway, if there are not sufficient reasons, as the success of AVS is usually determined by SI>5, which is the basis for calculating LI, it would be better to exclude cases with SI<5.
Table 1
The position of the words “systolic” and “diastolic” is the other way around.
Table 2
To understand the correlation between conventional LI and modified LI, the author might want to make a dot-plot diagram of the corresponding conventional LI and modified LI in rt APA, lt APA, and IHA respectively.
Table 3
The author should add a headline to it.
Figure 2.
As I mentioned, these analyses are not enough to support this theory. The scatter plot between the A/C values in Lt AdV divided by the total of those in Rt AdV and Lt AdV A/C and those in S-rt AdV divided by that could improve the validity of this theory. The comparison between the values in IVH and S-rt AdV is important as well. If this theory is correct, the values related to S-rt AdV will be higher than the others.
Author Response
Reviewer 2
The method is interesting as well as the results, however, the protocol to sample from the confluence with the right AdV is not practical. As the author mentioned in the discussion, we will not need to sample from the confluence if we have success to detect the right adrenal vein. They should have sampled from the point that might be around the right adrenal vein considering CT and analyzed it as well.
The theory is simple, but not seem to be persuasive. The concentration gradients of aldosterone and cortisol vary from site to site, partly due to the metabolic effects of the tissues. In the author’s theory, the point for sampling would not have had to be strictly just over one vertebra as long as it is at least above the right adrenal vein. The same might be true if blood is drawn in the left renal vein rather than the left adrenal vein, for example. Unlike the adrenal vein, as the inferior vena cava is large in diameter, its values are likely to be variable because of the mixture of blood flow from many tissues.
Considering those, rather than the values of Modified LI itself being important, the novelty could lie in the fact that this kind of method is expected to gather data, adding new insights to the AVS.
Thank you for your comment.
The purpose of this research was devised with the idea of a backup method for the conventional method, not a method to replace the conventional method.
The reason for setting the position of the catheter above one vertebra is to prevent it from being positioned below the right adrenal sinus.
Comments)
In the material and methods section, as primary aldosteronism is sometimes complicated by subclinical Cushing syndrome, the author should describe whether these patients were tested 1 mg dexamethasone suppression test, which could affect the result to analyze LI.
Blood ACTH and blood cortisol levels were measured in all cases, and dexamethasone suppression tests were performed in many cases.
Line 82
If the lateralization in these patients is diagnosed by using modified LI, which side will they be on?
There were both bilateral, right-sided, and left-sided cases. We did not evaluate these cases in which localization could not be clarified at this time.
Line 86-87
The sentence “patients whose blood pressure did not decrease to within the normal range (n=0)” is not needed.
Line90-91
Does the normal range mean ARR<200?
Yes, we added ”(<200)”
Line 91
What is the definition of adrenal lesions? Is the tumor over 10 mm?
We have not defined the size. Based on diagnosis by expert radiologists.
Line 96-97
The sentence “At preoperative CT, the adrenal lesions were detected in 30 patients (right: 9, 96 left: 17, bilateral: 4) and 14 patients in which no lesions were detected.” could be omitted being similar to the sentence in line 92-93.
Thank you. The sentence has been omitted.
Line 132
I think the correct sentence is “LI < 4.0 was diagnosed as IHA”. However, I can see the patient with LI >4 in the IHA group as an outlier in figure 3. How does the author diagnose this patient with IHA? According to the exclusion criteria, the author seems to consider the postoperative results for definition as well, especially in APAs.
I'm sorry. I don't understand what you mean.
Line 188
Considering the variation of SD in Tables, I recommend the author use median and quartiles and perform a Mann-Whitney test when checking the significance.
Mann-Whitney test is performed. It is written in the first line of Statistical Analysis section.
Line 191
Considering the values in Table 2, it seems to be a mistake for 1142.9 instead of 11429.9.
Thank you. We have been collected.
Line 208
The purpose to perform AVS is to find the unilateral lesion, meaning the specificity to detect it is important. The author might want to describe the values showing 100% specificity.
A 100% specificity of being right or left unilateral results in almost all bilateral diagnoses. Considering that this method is a supplementary diagnostic method for conventional methods, we believe that this evaluation method is sufficient.
Line 215-216
It is said that in 5 of the 44 cases in which catheterization of the right AdV was deemed technically successful, the SI was<5, suggesting unsuccessful catheterization. In general, I think it is a technical error if they cannot get the result that SI was >5. Does this sentence mean these patients have nonfunctional adrenal adenomas having the latent secretion of cortisol or some veins to the adrenal vein, affecting the AVS results? Anyway, if there are not sufficient reasons, as the success of AVS is usually determined by SI>5, which is the basis for calculating LI, it would be better to exclude cases with SI<5.
The clinical significance of the method devised this report is that we also wanted to examine whether it would be possible to back up cases in which the selection was unsuccessful in the blood collection data at a later date, in cases where the procedure was considered successful, so we included such cases. If it is better to omit it, omit it and analyze again.
Table 1
The position of the words “systolic” and “diastolic” is the other way around.
Thank you, we have collected them
Table 2
To understand the correlation between conventional LI and modified LI, the author might want to make a dot-plot diagram of the corresponding conventional LI and modified LI in rt APA, lt APA, and IHA respectively.
It is shown as a boxplot in Fig3.
Table 3
The author should add a headline to it.
Thank you we have added.
Figure 2.
As I mentioned, these analyses are not enough to support this theory. The scatter plot between the A/C values in Lt AdV divided by the total of those in Rt AdV and Lt AdV A/C and those in S-rt AdV divided by that could improve the validity of this theory. The comparison between the values in IVH and S-rt AdV is important as well. If this theory is correct, the values related to S-rt AdV will be higher than the others.
This Fig is to explain this hypothesis in an easy-to-understand manner, and in reality it will not be like this due to dilution and influx. However, it is hypothesized that it shows such a trend. In addition, this method is considered to be used as an auxiliary method.

Round 2
Reviewer 1 Report
The Authors have made an effort to address my criticisms.
However, most of my criticisms still remain, including far too small sample size, selection bias, lack of background evidence for using this sampling technique, use of ACTH that is known to be misleading etc., just to mention a few.
In addition, calculation of sensitivity and specificity in this type of studies that aim at improving the diagnosis on an individual basis is inappropriate. The Authors should refer to the literature in this regards, for example the manuscript by John Attia (Aust Prescr 2003;26:111–3.
Author Response
Reviewer 1
The Authors have made an effort to address my criticisms.
However, most of my criticisms still remain, including far too small sample size, selection bias, lack of background evidence for using this sampling technique, use of ACTH that is known to be misleading etc., just to mention a few.
In addition, calculation of sensitivity and specificity in this type of studies that aim at improving the diagnosis on an individual basis is inappropriate. The Authors should refer to the literature in this regards, for example the manuscript by John Attia (Aust Prescr 2003;26:111–3.
Thank you for your comment.
Problems with a small number of cases have been added to the limitation as following sentence.
“The fourth limitation was the small number of patients. This study needs to increase the number of cases and evaluate in the future.”
There was no bias in patient selection and Indications for AVS. As described in materials and methods with reference to guidelines.
There is no evidence that the use of ACTH improves diagnostic performance, but guidelines recommend its use because it improves the accuracy of the selectivity index
The manuscript has been revised based on the likelihood ratios as suggested (Results in Abstract, line5-8 in Statistical analysis, line10-13 in Results, Figure legend of Figure5, and Table3).

Reviewer 2 Report
The author revised some parts of the manuscript; however, some of the answers do not satisfy me. I added comments to the author’s replies. I would like the author to make minor revisions to see whether the author answers my comments clearly and sincerely.
Comments to Author’s replies)
First of all, the author should add the paragraph number that should help to give comments.
1. We have not defined the size. Based on diagnosis by expert radiologists.
The author should add this diagnostic criterion to the manuscript, which could help readers who want to do research about adrenal lesions.
2. I'm sorry. I don't understand what you mean.
I am talking about the sentence on page 8 “LI ³ 4.0 was diagnosed as IHA” that looks wrong. The correct sentence could be “LI < 4.0 was diagnosed as IHA”. Considering this, looking at Figure 3, we can see the dot showing over 10 in LI in IHA, which should be APA in this diagnostic criterion because the LI was > 4. How did the author diagnose this case with IHA?
3. Mann-Whitney test is performed. It is written in the first line of Statistical Analysis section.
As I have already mentioned, the author should change the mean ARR, PAC, and PRA to the median if you use Mann-Whitney test following my advice because these results do not look normally distributed. However, if the author does not want to, I understand that the author thinks these values are in the normal distribution, which is not generally wrong. It depends on the author.
By the way, P values in Figure 3 and Figure 4 are not changed in the revised manuscript compared to the past one. Were the P values with Mann-Whitney test the same as those with t-test in Figure 3 and Figure 4?
4. The clinical significance of the method devised this report is that we also wanted to examine whether it would be possible to back up cases in which the selection was unsuccessful in the blood collection data at a later date, in cases where the procedure was considered successful, so we included such cases. If it is better to omit it, omit it and analyze again.
I understood what the author means. Then I recommend the author to omit the sentence “with technical success” in the sentence written by “Finally, this study included 44 patients (21 men, 23 women; mean age 56.0 years, range 28-73 years) with technical success of AVS, and all patients who were diagnosed with UAPA by AVS underwent surgery, and the ARR decreased to the normal range (<200).”
5. This Fig is to explain this hypothesis in an easy-to-understand manner, and in reality it will not be like this due to dilution and influx. However, it is hypothesized that it shows such a trend. In addition, this method is considered to be used as an auxiliary method.
If the author admits that the value of S-rt AdV A/C is not like the ideal one due to dilution and influx, this result especially about the values of modified LI could not have external validity. Saying there is a report supporting this hypothesis, the author should cite the report. If not, the author should have a comment about this in the limitation section.
Author Response
Reviewer 2
The author revised some parts of the manuscript; however, some of the answers do not satisfy me. I added comments to the author’s replies. I would like the author to make minor revisions to see whether the author answers my comments clearly and sincerely.
Thank you for your comment.
Comments to Author’s replies)
First of all, the author should add the paragraph number that should help to give comments.
- We have not defined the size. Based on diagnosis by expert radiologists.
The author should add this diagnostic criterion to the manuscript, which could help readers who want to do research about adrenal lesions.
Line18 1st paragraph in materials and methods
We added the diagnostic criterion as following.
“Preoperative CT showed adrenal lesions in 30 cases and no lesions in 14 cases which diagnosed by radiologist.”
- I'm sorry. I don't understand what you mean.
I am talking about the sentence on page 8 “LI ³ 4.0 was diagnosed as IHA” that looks wrong. The correct sentence could be “LI < 4.0 was diagnosed as IHA”. Considering this, looking at Figure 3, we can see the dot showing over 10 in LI in IHA, which should be APA in this diagnostic criterion because the LI was > 4. How did the author diagnose this case with IHA?
Thank you. I corrected “LI < 4.0 was diagnosed as IHA”
As you pointed out, it is strange that the IHA in Fig3 has a value of 10 or more. Therefore, when I reviewed it, 3.12 was entered as 13.12 by mistake, so I corrected Fig3. This mistake is only an error when creating a graph, and other results such as statistics can be left as they are. Thank you.
- Mann-Whitney test is performed. It is written in the first line of Statistical Analysis section.
As I have already mentioned, the author should change the mean ARR, PAC, and PRA to the median if you use Mann-Whitney test following my advice because these results do not look normally distributed. However, if the author does not want to, I understand that the author thinks these values are in the normal distribution, which is not generally wrong. It depends on the author.
By the way, P values in Figure 3 and Figure 4 are not changed in the revised manuscript compared to the past one. Were the P values with Mann-Whitney test the same as those with t-test in Figure 3 and Figure 4?
Your opinion is correct. However, we consider these data to be normally distributions and we take the mean, but we can change that if it is needed.
Statistical analysis was performed using Mann-Whitney test from the beginning, so the result doesn't change
- The clinical significance of the method devised this report is that we also wanted to examine whether it would be possible to back up cases in which the selection was unsuccessful in the blood collection data at a later date, in cases where the procedure was considered successful, so we included such cases. If it is better to omit it, omit it and analyze again.
I understood what the author means. Then I recommend the author to omit the sentence “with technical success” in the sentence written by “Finally, this study included 44 patients (21 men, 23 women; mean age 56.0 years, range 28-73 years) with technical success of AVS, and all patients who were diagnosed with UAPA by AVS underwent surgery, and the ARR decreased to the normal range (<200).”
Thank you for your understanding. I omitted “with technical success”
- This Fig is to explain this hypothesis in an easy-to-understand manner, and in reality it will not be like this due to dilution and influx. However, it is hypothesized that it shows such a trend. In addition, this method is considered to be used as an auxiliary method.
If the author admits that the value of S-rt AdV A/C is not like the ideal one due to dilution and influx, this result especially about the values of modified LI could not have external validity. Saying there is a report supporting this hypothesis, the author should cite the report. If not, the author should have a comment about this in the limitation section.
I added the following sentence in the limitation section.
“The fourth limitation was this study is based on the hypothesis shown in Fig. 2, but there is no report to support it. But it also matters of course, as the study aims to demonstrate a new methodology. We believe that it will show the tendency shown in the hypothesis.”
The purpose of this study is to simplify AVS, which is often difficult to complete, and to enable it to be performed in many institutions with no specialist. We believe that it will benefit patients to inform widely this method as an adjunct rather than replacing existing methods.
Thank you for your advice!

Round 3
Reviewer 2 Report
The author answered all my comments. I think the manuscript is enough to be published.
Author Response
Thank you for your advice